# The Prognostic and Predictive Value of Human Gastrointestinal Microbiome and Exosomal mRNA Expression of PD-L1 and IFNγ for Immune Checkpoint Inhibitors Response in Metastatic Melanoma Patients: PROTOCOL TRIAL

**DOI:** 10.3390/biomedicines11072016

**Published:** 2023-07-18

**Authors:** Ana Erman, Marija Ignjatović, Katja Leskovšek, Simona Miceska, Urša Lampreht Tratar, Maša Bošnjak, Veronika Kloboves Prevodnik, Maja Čemažar, Lidija Kandolf Sekulovič, Gorazd Avguštin, Janja Ocvirk, Tanja Mesti

**Affiliations:** 1Department of Medical Oncology, Institute of Oncology Ljubljana, Zaloška Cesta 2, 1000 Ljubljana, Slovenia; aerman@onko-i.si (A.E.); mignjatovic@onko-i.si (M.I.); kleskovsek@onko-i.si (K.L.); jocvirk@onko-i.si (J.O.); 2Medical Faculty, University of Ljubljana, Kongresni Trg 12, 1000 Ljubljana, Slovenia; 3Department of Cytopathology, Institute of Oncology Ljubljana, Zaloška Cesta 2, 1000 Ljubljana, Slovenia; smiceska@onko-i.si (S.M.); vkloboves@onko-i.si (V.K.P.); 4Department of Experimental Oncology, Institute of Oncology Ljubljana, Zaloška Cesta 2, 1000 Ljubljana, Slovenia; ulampreht@onko-i.si (U.L.T.); mbosnjak@onko-i.si (M.B.); mcemazar@onko-i.si (M.Č.); 5The Military Medical Academy, 11000 Beograd, Serbia; lkandolfsekulovic@gmail.com; 6Biotechnical Faculty, University of Ljubljana, Kongresni Trg 12, 1000 Ljubljana, Slovenia; gorazd.avgustin@bf.uni-lj.si

**Keywords:** gastrointestinal microbiome, mRNA expression of PD-L1 and IFNγ, immune checkpoint inhibitors, metastatic melanoma, predictive and prognostic biomarkers, immune-related adverse events

## Abstract

Background: Immunotherapy has been successful in treating advanced melanoma, but a large proportion of patients do not respond to the treatment with immune checkpoint inhibitors (ICIs). Preclinical and small cohort studies suggest gastrointestinal microbiome composition and exosomal mRNA expression of PD-L1 and IFNγ from the primary tumor, stool and body fluids as potential biomarkers for response. Methods: Patients treated with immune checkpoint inhibitors as a first line treatment for metastatic melanoma are recruted to this prospective study. Stool samples are submitted before the start of treatment, at the 12th (+/−2) week and 28th (+/−2) week, and at the occurrence of event (suspected disease progression/hyperprogression, immune-related adverse event (irAE), deterioration). Peripheral venous blood samples are taken additionally at the same time points for cytologic and molecular tests. Histological material from the tumor tissue is obtained before the start of immunotherapy treatment. Primary objectives are to determine whether the human gastrointestinal microbiome (bacterial and viral) and the exosomal mRNA expression of PD-L1 and IFNγ and its dynamics predicts the response to treatment with PD-1 and CTLA-4 inhibitors and its association with the occurrence of irAE. The response is evaluated radiologically with imaging methods in accordance with the irRECIST criteria. Conclusions: This is the first study to combine and investigate multiple potential predictive and prognostic biomarkers and their dynamics in first line ICI in metastatic melanoma patients.

## 1. Introduction

Melanoma is a dangerous form of skin cancer, accounting for around 5% of all skin cancers and responsible for more than 90% of skin cancer deaths [1,2,3]. Although the survival rate is improving, the prognosis depends on the stage at diagnosis [4,5,6,7]. In the last decade, immunotherapy has achieved the greatest advance in the treatment of disseminated melanoma. The median overall survival rate was greatly extended, thus giving hope to many patients. However, only up to 50% of patients respond to immunotherapy treatment. Severe immune-related adverse events (irAE) may occur during treatment, and some patients have no or minimal benefit from immunotherapy treatment due to primary or acquired resistance [8]. There is a need to implement biological markers in clinical practice to improve personalized treatment and to predict response to treatment [7]. The absence of clinically validated predictive biomarkers is one of the biggest causes of the unpredictable effect of immunotherapy [9]. Based on the existing literature, the microbiome, PD-1/PD-L1 (programmed death-ligand 1/ programmed death 1) signal pathway and Interferon-gamma (IFN-γ) are suggested as possible biomarkers, but further studies are needed to obtain relevant clinical information on larger cohorts.

### 1.1. Immunotherapy with Immune Checkpoint Inhibitors in the Treatment of Malignant Melanoma

The immune system plays an important role in carcinogenesis. The suppression of immune system creates a favorable tumor microenvironment that promotes the growth and development of cancer cells. In recent years, a breakthrough in oncology has been achieved by immunotherapy with immune checkpoint inhibitors (ICIs). ICIs are monoclonal antibodies that, by binding to PD-L1/PD-1 and CTLA-4 (cytotoxic T-lymphocyte-associated protein 4), enable immune cells to act against tumors [10]. *PD-1* is a transmembrane receptor on various cells of the immune system (T and B lymphocytes, macrophages, natural killer cells and dendritic cells). Transmembrane glycoprotein *PD-L1* is found on immune cells, such as macrophages, some activated T and B lymphocytes, dendritic and epithelial cells, and is also present on tumor cells, where it enables so-called “immune escape”. The binding of PD-1 to PD-L1 initiates a negative regulatory loop of T lymphocytes and other immune cells, thereby enabling the proliferation of cancer cells. According to recent research, PD-L1 can also independently activate proliferative tumor pathways (PI3K/AKT, MAPK, JAK/STAT, Hedgehog and others) [11]. CTLA-4 appears as a protein receptor in regulatory T lymphocytes and, by binding to the B7 protein on antigen-presenting cells, initiates the inhibition of the immune response [5].

Immunotherapy with PD-1/PD-L1 and CTLA-4 inhibitors in the treatment of patients with metastatic malignant melanoma has greatly improved prognosis and survival. Life expectancy has increased from less than 12 months to more than 40 months. Immunotherapeutics from the group of PD-1 inhibitors (nivolumab, pembrolizumab), a combination of two-tier immunotherapy of PD-1 and CTLA-4 inhibitors (nivolumab and ipilimumab) and BRAF inhibitors (vemurafenib, dabrafenib)—and MEK (cobimetinib, trametinib) signaling pathways in the case of BRAF V600 mutation—are used as the first line treatment for metastatic melanoma according to ASCO and NCCN guidelines [12].

Unlike BRAF and MEK inhibitors, which inhibit signaling through the RAS–RAF–MEK signaling pathway, the action of immunotherapy is much more complex and therefore more unpredictable. In approximately 10% of patients, the response to immunotherapy treatment manifests as pseudoprogression, with a discrepancy between the clinical and radiological response and the actual response due to a transiently violent inflammatory response. Up to 30% of patients respond to immunotherapy treatment with hyperprogression, where, for as yet unknown reasons, the cancer progresses rapidly. More than a third of patients have irAE when treated with ICI, but regardless, treatment interruption or discontinuation has a better response and longer progression free survival [13,14,15].

### 1.2. Human Gastrointestinal Microbiome, PD-1/PD-L1 Signal Pathway and Interferon-Gamma (IFN-γ) as Prognostic and Predictive Biomarkers

#### 1.2.1. Human Microbiome

The human microbiome is a genetic set of all microbes in the human organism including bacteria, viruses, fungi and protozoa. It consists of approximately 100 trillion microorganisms that encode more than three million genes and thousands of metabolites [16]. The microbiome affects metabolic homeostasis, metabolism and neurological development, and also plays an important role in the immune response [17,18]. An imbalance between the microbiome and its host is believed to have an impact on the occurrence of many diseases, such as cancer and autoimmune, metabolic and infectious diseases. The composition of the microbiome is influenced by environmental factors during and after birth, childhood and adulthood, and it also changes with diet and nutrition [19].

Recently, techniques for more precise analysis of the microbial genome and the interactions between the microbiome and its host have been rapidly developed. DNA sequencing technology, together with advances in bioinformatics, enables the amplification and sequencing of target microbiological DNA regions, followed by statistical analysis and grouping of microorganisms according to the collected databases. When studying the bacterial genome, we use the target of the 16S ribosomal gene (16S rDNA), a hypervariable sequence, on the basis of which we distinguish individual bacterial genera. Metagenomic sequencing is also in use, which allows a more precise insight into the so-called microbial metagenome (set of all microbial genes from an ecosystem) and simultaneous identification of the composition of the microbiome down to the level of bacterial species, as well as insight into the composition of the so-called virome, i.e., part of the microbiome that includes bacteriophages and eukaryotic viruses [16].

#### 1.2.2. Immunobiology and the Human Gastrointestinal Microbiome

The largest concentration of microbes and lymphoid tissue is in the gastrointestinal tract (GIT), where the immune system and the microbiome are in symbiosis [18]. The microbiome participates in the production of cytokines, regulates the T cell response and maintains homeostasis [19]. It has been demonstrated that changes in the composition and density of the GIT microbiota locally affect the immune response, but the question of the effect on the immune and inflammatory response at the systemic level arises [20]. The influence of the microbiome on systemic immunity is mainly attributed to the activation of specific regulatory T cells and the production of interleukins (interleukin 9 and others) [21].

#### 1.2.3. Immunotherapy with Immune Checkpoint Inhibitors and the Microbiome

Despite the great success of the treatment of metastatic malignant melanoma with immunotherapy, some patients do not respond to the treatment, or develop severe adverse side effects during the treatment that prompt discontinuation of the treatment. Some biomarkers of the tumor microenvironment that predict the response to immunotherapy treatment are already known, such as tumor mutational burden (TMB) and microsatellite stability (MSS). Some of the studies carried out so far suggest the GIT microbiome and its cytokines as potential biomarkers of carcinogenesis, and that they could also serve as predictive biomarkers of response to treatment with PD-1 and CTLA-4 inhibitors. The impact of dysbiosis due to antibiotic use on the efficacy of treatment with PD-1 and CTLA-4 inhibitors has been reported, as has the improvement of responses to treatment with immune checkpoint inhibitors in patients after fecal microbiome transplantation (FMT) [8].

#### 1.2.4. PD-1/PD-L1 Signal Pathway

The PD-1/PD-L1 signaling pathway is an important mechanism for immune escape and tumor progression. Recent research has shown that patients with overexpression of PD-L1 in their tumor tissue benefit more from treatment with programmed cell death protein 1 (PD-1) and PD-L1 inhibitors [22,23]. Programmed cell death receptor 1 (PD-L1) on the surface of tumor cells inhibits the antitumor activity of T lymphocytes by binding to their PD-1 receptor and causes immunosuppression [24,25]. PD-L1 is a promising predictive biomarker for predicting response to immunotherapy treatment. An alternative to membrane-bound PD-L1 is exosomal PD-L1 (ExoPD-L1), secreted from tumor cells, whose expression correlates with membrane-bound PD-L1 on the surface of the parental tumor cell [24,25]. The predictive role of PD-L1 is controversial, as around 10% of patients who do not express PD-L1 in tumor tissue also respond to immunotherapy. This is probably related to the immune microenvironment of the tumor, where PD-L1 is also expressed on immune cells (on helper T cells—CD4+, killer T cells—CD8+, regulatory T lymphocytes—Treg, B lymphocytes, killer cells—CD56+ and monocytes -macrophage cells) and contributes to antitumor immunity.

PD-L1 expression is dynamic and variable with patient and tumor heterogeneity and may depend on several factors, including prior therapies and the presence of tumor-infiltrating immune cells [12]. The expression can vary from the tumor margin to the core, between the primary tumor and metastases, and change dynamically during disease progression. Consequently, a biopsy from tumor tissue at a given time is not representative enough to determine the PD-L1 status of a tumor. Given the lack of detection of PD-L1 by conventional biopsy and IHC, research is focusing on the expression of circulating PD-L1 in serum, plasma, circulating tumor cells, and exosomes, which are minimally invasive and obtain real-time detection for a more accurate representation of heterogeneous PD-L1 expression [23,26].

The gut microbiome and its interactions with the tumor microenvironment, up- and down-regulating a beneficial immune response, could be a potential predictive biomarker for immunotherapy efficiency. The data has shown that beneficial gut microbiota in responders enhance the activation and expression of CD cells and promote CD cell activation, as opposed to non-responders with unfavorable gut microbiota and a higher frequency of Tregs [27].

#### 1.2.5. Interferon Gamma

Interferon Gamma (IFNγ) is a cytokine with antitumor and immunomodulatory activity [28].

It is formed by the cells of the immune system—cytotoxic T lymphocytes (CD8+TLy), helper T cells (CD4+TLy) and natural killer cells (NK cells) [29]. IFNγ acts by activating the JAK/STAT signaling pathway, which modulates the transcription of several genes [30]. Some of the end products of these genes inhibit the growth of tumor cells, while others contribute to it. The anti-tumor activity of IFNγ is manifested by the activation of antigen-presenting cells, the arrest of the cell cycle in the G1 phase, the stimulation of cell ischemia and apoptosis. Pro-tumor activity is caused by T cell suppression and inhibition of NK cell activity, stimulation of programmed cell death ligand 1 (PDL1) expression on tumor cells and stimulation of angiogenesis and tumorigenesis [28].

Recent research has shown that IFNγ reflects the state of response to immunotherapy. Higher concentrations of IFNγ before starting the treatment are associated with a better response to treatment [30,31,32]. Reduced expression of genes encoding the synthesis of IFNγ itself or its receptor, as well as genes that are part of the JAK/STAT pathway, results in its reduced activity and, consequently, resistance to treatment with immune checkpoint inhibitors [33,34,35,36,37]. Some recent studies have also shown that the gut microbiome modulates IFNγ response and plays a role in opportunistic infections and autoimmune diseases [38,39].

#### 1.2.6. Rationale for the Study

The specific composition and biodiversity of the gastrointestinal microbiome, exosomal mRNA expression of PD-L1 and IFNγ and dynamics predict the response to immunotherapy treatment with immune check point inhibitors and could serve as prognostic and predictive markers.

The studies published so far have been carried out on animal models or a small number of subjects or samples. We attribute the conflicting results of the studies conducted so far to this. The importance of intestinal flora in health or pathology and the influence and connection of inflammation and immune cells is a developing medical topic. It is for this reason that many studies are based on small cohorts and a small number of samples, especially in the field of oncology. Our study is pioneering precisely in this respect, as it is composed thoroughly, with a clear hypothesis, on a large cohort and number of samples. Samples will be taken at three different treatment points and at any event such as hyperprogression, pseudoprogression or immune-related side events. We will study the dynamics of changes in the diversity of the GIT microbiome and the systemic immunological response during the treatment of metastatic melanoma with immunotherapy.

The study is first prospective study that simultaneously determines the expression of IFNγ and PD-L1 in tumor tissue and blood at different time points. If we demonstrate that there is a correlation in the expression of IFNγ and PD-L1 in the tumor tissue and the blood, IFNγ could be obtained from the patient’s peripheral blood by a minimally invasive method.

## 2. Methods and Design

### 2.1. Study Design

The study is cohort-prospective, non-randomized and non-interventional (Figure 1). It will assess the association between the predominant composition of the human gut microbiome and the response to immunotherapy treatment with PD-1 and CTLA-4 inhibitors in metastatic malignant melanoma. It will also observe whether specific biomarkers (host immune cells population, PD-L1 and IFNγ expression) correlate with the predominant composition of the gut microbiome and the response to treatment with immunotherapy.

Patients with metastatic malignant melanoma who will be treated with immunotherapy in the first line in the three-year period from October 2022 to October 2025 will be included in the research. One-hundred-and-fifty patients will be included. Consenting patients will be asked to provide clinical data and donate biological samples before, during and after completing their treatment. Patient data will be anonymized—date of birth, gender, date of diagnosis, performance status at initiation of treatment, liver enzymes, tumor markers—LDH, S100, localization, BRAF status, treatment type, response to immunotherapy, treatment time, immune-related adverse events of treatment.

### 2.2. Study Objectives

#### 2.2.1. Primary Objectives

To determine the relationship between the predominant composition of the human gastrointestinal microbiome and the objective response to treatment with ICIs in patients with metastatic malignant melanoma.To determine whether the dynamics of exosomal mRNA expression for PD-L1 and PD-L1 on the surface of immune cells is related to the response to immunotherapy treatment and has predictive value.To determine whether the exosomal mRNA expression for IFNγ is related to the response to immunotherapy treatment and has a predictive value.

#### 2.2.2. Secondary Objectives

To determine whether a specific composition of the microbiome is associated with the occurrence of immune-related side effects in treatment with ICIS.To determine the effects of ICI on the gut microbiota over different time points.

#### 2.2.3. Exploratory Objectives

To determine if there is a relationship between a certain composition of the microbiome and the cell population (CD3+, CD4+, CD8+, the ratio between CD4+ and CD8+, macrophages) in the peripheral blood.Whether patients who develop immune-related side effects when treated with immunotherapy have a better response to it and longer survival without disease progression compared to patients who do not develop immune-related side effects.

### 2.3. Recruitment and Eligibility of Patients

#### 2.3.1. Study Population

The study will recruit patients with metastatic melanoma (TNM classification, stage M) receiving first-line therapy with immune checkpoint inhibitors (ICIs). They will be treated with PD-1 inhibitors (pembrolizumab, nivolumab) or with a combination of PD-1 and CTLA-4 inhibitors (dual immunotherapy ipilimumab/nivolumab).

According to the phase III clinical research protocols that monitored OS and PFS in the treatment of metastatic malignant melanoma with ICIs, we will continue the treatment even if there is radiologically proven progress and the absence of clinical and biochemical signs of disease progression with sufficient suspicion of pseudoprogression.

#### 2.3.2. Inclusion Criteria

Age over 18 years;Cytologically or histologically verified malignant melanoma;Stage IIID unresectable/IV according to AJCC classification (8th edition, 2018);Performance status according to WHO 0–2 (ECOG criteria);1st line of systemic treatment with immunotherapy (nivolumab, ipi/nivo, pembrolizumab);Triple CT (head, thorax, abdomen)/PET CT done within 4 weeks before the first application;Signed consent to participate in clinical research.

#### 2.3.3. Exclusion Criteria

Previously treated melanoma with systemic therapy;Capacity status according to WHO 3–4 (ECOG criteria);Contraindications for immunotherapy treatment (known deficiency of the immune system or active immunosuppressive treatment or active autoimmune disease requiring treatment);Other malignant diseases (except cured basal cell carcinoma and squamous cell carcinoma).

#### 2.3.4. Recruitment Process

Only eligible patients with a signed consent form will participate in the study.

#### 2.3.5. Collaborating Centers and Departments

-Department of Medical Oncology, Institute of Oncology Ljubljana, Slovenia;-Department of Experimental Oncology, Institute of Oncology Ljubljana, Slovenia;-Department of Cytopathology, Institute of Oncology Ljubljana, Slovenia;-Biotechnical Faculty, University of Ljubljana, Slovenia;-The Military Medical Academy, Belgrade, Serbia.

#### 2.3.6. Assessments during the Study

Patients will be followed up at their routine visits at the clinic. The follow-up will be scheduled at 3- or 4-week intervals, depending on the ICIs treatment regimen, when routine examination and laboratory test will be taken. Study-specific data will be collected at three different time points: up to 4 weeks before the start of treatment, at the 12th (+/−2 weeks) and 28th week (+/−2 weeks). Study-specific data will be taken additionally, if suspected disease progression and the appearance of immune-related adverse events or if indicated by the oncologist (Table 1).

Patients will be followed up regularly for the time of the treatment for the first 12 months. The study will be prematurely terminated in an event occurrence defined as disease progression/hyperprogression; by the physician’s choice in severe immune-related adverse events; when high doses of systemic corticosteroid (dexamethasone 4 mg/day or equivalent or higher) or prolonged antibiotic treatment are present; or by the patient’s choice. Before each stool sample donation, the patient will fulfill a questionnaire regarding eating habits, alcohol and tobacco consumption and medications. In the case of antibiotic treatment, the stool sample will be donated after 3 weeks of the last antibiotic dose taken.

#### 2.3.7. Immune-Related Adverse Events (IrAEs) Data Collection and Evaluation

Immune-related adverse events (IrAEs) will be reported at each follow-up visit. IrAEs will be evaluated based on the established and standardized grading system Common Terminology Criteria for Adverse Events (CTCAE) developed by the National Cancer Institute. This system grades adverse events based on their severity, ranging from grade 1 (mild) to grade 5 (death). Comprehensive and standardized data on irAEs will be obtained from the patient on each clinical visit, including relevant patient characteristics, timing of irAE onset, clinical manifestations, laboratory findings, imaging results and interventions undertaken. The biological samples from the patients will be taken if a grade 2 event or higher is noted by a physician. The biological samples will be taken within 2 weeks of the IrAE first reported. Results will be given as a percentage of incidence and the severity of IrAEs based on grading will also be reported.

### 2.4. Biological Sample Collection

#### 2.4.1. Collection of Peripheral Venous Blood Samples

Peripheral venous blood samples will be taken during a regular visit to OI Ljubljana up to 4 weeks before the start of treatment, the 12th (+/−2) and the 28th (+/−2) week, in the event of suspected disease progression/hyperprogression and in the event of the occurrence of immunologically unwanted side effects (colitis or other immune-related side effects).

During regular blood sampling, a sample of blood will be taken from the examinee for additional tests—a total of 10 mL, of which 5 mL will be used for cytologic tests (specific immune host cell concentration) and another 5 mL of the venous blood will be used for molecular tests (messenger RNA for PD-L1 and IFNγ expression).

#### 2.4.2. Collection of Tumour Samples

Histological material from the tumor tissue will be obtained before the start of immunotherapy treatment. At the Department of Pathology of the Ljubljana Oncology Institute, multitumor blocks and cut material for molecular examinations will be prepared from histological samples.

#### 2.4.3. Collection of Stool Sample

The patient will submit a stool sample during regular defecation, before the start of treatment, at the 12th (+/−2) and 28th (+/−2) week, in the event of suspected disease progression/hyperprogression and in the event of the occurrence of immune adverse side effects (colitis or other irAE). According to the received instructions, they will send it by post in the attached envelope together with the completed questionnaire. In the case of antibiotic treatment, the collection of the stool sample will be postponed and taken 3 weeks after the antibiotic treatment has been completed. The samples stored in the OMNIgene transport medium will be sent to the Faculty of Biotechnology of the University of Ljubljana, where microbiome analysis will take place.

### 2.5. Sample Analysis

#### 2.5.1. Gut Microbiome, Metagenome and Virome Analysis

The samples stored in the OMNIgene transport medium will be sent to the Faculty of Biotechnology of the University of Ljubljana, where the microbiome analysis will take place. After the successful isolation of total microbial DNA from thawed stool samples using the appropriate protocols (with a special procedure that will include the removal of non-viral particles by centrifugation and filtration through membrane filters with 0.2 μm pores), we will also obtain viral DNA from a smaller, selected set of samples, to approach the analysis of the microbiome, metagenome and virome. We will analyze the structure and changes in the intestinal microbiome with the so-called by amplicon sequencing of 16S rRNA—we will use the technology Illumina 16S/ITS Nextera two-step PCR &MiSeq 2 × 300. The variable regions of genes for 16S rRNA will be analyzed with appropriate bioinformatic and biostatistical analysis: UPARSE pipeline, qualitative filtering with parameter settings that will be optimized after analyzing the so-called “mock” pattern. The identification of operational taxonomic units (OTE) with the “usearch global” function in Silva NR SSU and LTP SSU databases, taxonomic classification in OTE with Wang’s method built into the software package Mothur v 1.35. Taxonomy, OTE tables and phylogenetic trees will be made with the “R” package phyloseq for statistical analysis and graphics.

We will perform metagenomic sequencing by constructing appropriate Nextera libraries and sequencing with the Illumina NovaSeq6000 2 × 150 bp platform. We expect 20 million readings per sample. We will process the acquired sequences accordingly (quality filtering with FastQC, Trimmomatric), remove human sequences with bbmap, and analyze the sequences with established metagenomic software packages (pipelines, e.g., Assnake). We will obtain taxonomic and functional profiles of processed metagenomes with MeatPhIAn3 and HUMAnN2 tools and the KEGG database.

Analysis of virome composition will be performed by metagenomic sequencing, for which we will use Illumina NovaSeq technology, TruSeqlibrary, 2 × 150 bp, and we will obtain up to 5 mio. of readings per sample. We will process the acquired sequences accordingly (quality filtering with FASTQC, Trimmomatic) and assemble contigs (with the metaSpades assembler). We will keep only those that we will recognize with the appropriate tools (Virsorter) or will be recognized in the RefSeq Virus database with Blast, and we will remove all bacterial chromosomal DNA contigs.

#### 2.5.2. Tumor Tissue and Peripheral Blood Analysis

The MagMAX™ FFPE DNA/RNA Ultra Kit will be used to isolate RNA from paraffin. Tissue slices (40 µm) will be prepared from the tumor blocks in the Department of Pathology, after which further analysis will take place in the Department of Experimental Oncology. Initially, the paraffin will be dissolved with the addition of xylene, by centrifugation and heating the samples to 50 °C. This will be followed by rinsing with 100% ethanol and drying the sample with a fixed tissue. The tissue sample will be incubated with protease buffer for 1 h at 50 °C and 1 h at 90 °C. RNA buffer will be added to bind the RNA to the magnetic beads. To remove DNA, a DNAse solution will be added after washing. This will be followed by repeated washing and centrifugation with the elution solution until the RNA supernatant is obtained. The sample will be stored at −80 °C until analysis. Sample analysis will be performed by RT-qPCR.The Plasma/Serum Circulating and Exosomal RNA Purification Kit (Slurry Format (Norgen Biotek Corp., Ontario, Canada)) will be used to isolate RNA from plasma. The analysis will take place on 5 mL of venous blood samples. Blood plasma will be prepared from them by centrifugation. To bind the RNA to the pellets, a 2 mL plasma sample will be incubated with lysis buffer at 60 °C. Then, after adding 100% ethanol, it will be centrifuged. For greater purity, the process will be repeated twice. The ultracentrifugation process will bind the mixture to the column. This will be followed by washing the sample and adding DNase to remove the DNA. The sample will be processed twice more by washing and ultracentrifugation to obtain the supernatant with RNA, which will be stored at −80 °C. RNA concentration will be determined spectrophotometrically with a Qubit fluorometer (Applied Biosystems, Massachusetts, USA). Analysis of the samples is expected to be done by ddPCR.PD-L1 expression on CD4+, CD8+ T lymphocytes and macrophages will be determined using the flow cytometric method. Antibody panels for 7-color flow cytometric measurements (antibodies from BD Biosciences (San Jose, CA, USA), BioLegend (San Diego, CA, USA) and Beckma Coulter (Brea, CA, USA)) will be used. Samples will be measured using a FACSCanto 10 flow cytometer (BD Biosciences). FACSDiva software (BD Biosciences) will be used to analyze flow cytometry results.Serum interferon gamma concentration: Quantikine human IFNγ ELISA kit (R&D Systems™, Minneapolis, USA) will be used to determine serum interferon gamma concentration.For the qRT-PCR assay, SYBR Green chemistry will be used, which enables relative, as well as absolute, quantification of the target sequence in the investigated sample. Relative quantification will be used to determine the expression level of the transgene in the paraffin samples. Additionally, absolute quantification will be used to determine the target sequences in the plasma of the patients using ddPCR. For both relative and absolute quantification, we will use specific set of primers (PrimeTime qPCR Primers, IDT Technologies, Coralville, IA, USA) that do not amplify any other known dsDNA sequence, and are specific for the IFNγ sequence and CD274 (PD-L1) sequence (primer details are in Table 2).Sample preparation for flow-cytometric measurement will be carried out as previously described by our group [38,39]. Used antibodies (Table 3) will be divided into 2 separate tubes according to the analyzed immune cells, and half a million cells per 100 μL will be used for each tube. Immune cells will be gated according to their immunophenotype: T cells (CD3+); the helper T cell subset (CD4+); cytotoxic T cell subset (CD8+); and macrophages (CD11b+CD14+), as well as their M1-like (CD86+) and M2-like (CD206+) subsets. Percentages of T cells and macrophages will be calculated as a ratio per CD45+. Percentages of CD4+ and CD8+ T cell subsets will be given as a percentage of CD3+, while M1-like and M2-like macrophage subsets will be given as a percentage per all macrophages. The expression of PD-1 and PD-L1 will be analyzed on both T cells and macrophages, as well as on their corresponding subsets. The correlation of the analyzed immune cells and their subsets with patient PFS and OS will be also calculated. Survival analysis will be based on an X-year patient follow-up. Flow-cytometric data will be acquired with a 10-color BD FACSCanto™ II Flow Cytometer and FACSDiva 8.0.2 software (BD Bioscience, USA). FSC files will be analyzed using FlowJo v10.8 1 (BD Biosciences, USA).

#### 2.5.3. Imaging Assessments

Patients will undergo diagnostic imaging (CT of the head, chest and abdomen or PET-CT) up to 4 weeks before the start of treatment, the 12th (+/−2) and 28th (+/−2) weeks after the start of treatment and at the occurrence of the event (in case of suspicion of disease progression, the occurrence of immune-related side effects, deterioration of the condition, etc.).

The response to the treatment will be evaluated radiologically with imaging methods in accordance with the irRECIST criteria (criteria for evaluating the immune response in solid tumors). The irRECIST criteria divide the response to treatment into individual groups: complete response (CR, complete response); partial response (PR, partial response); stagnation (SD, stable disease); and progression (PD, progressive disease). Pseudoprogression is defined as a transient radiological progression of the disease without a clinical picture and a gradual reduction of the burden of the underlying disease.

##### Interim Analysis

An interim analysis of data will be made of the first 20 patients. The results will be used to amend the protocol if necessary.

#### 2.5.4. Statistical Methods

The gastrointestinal microbiome, PD-L1 and Interferon Gamma expression will be correlated with one year of PFS as the primary outcome of the study. An interim analysis will be made after the first 20 included patients.

We will analyze the 16S rRNA sequences with the appropriate software tools from the Mothur software package, and we will perform analysis of molecular variance (AMOVA) and principal coordinate analysis (PCoA). For the statistical analysis of the differences in the representation of the sequences of individual bacterial groups in the studied microbiomes, we will use the DeSeq 2 package from the R programming environment. The association between phylogenetic diversity, taxonomic units, immune cells and the objective response to treatment will be determined using the Wilcoxon test. To compare values at different time points, we will use the paired *t*-test or corresponding non-parametric alternative. When comparing several groups at the same time, we will use the Kruskal–Wallis test. All differences will be considered statistically significant when *p* < 0.05.

The Kaplan–Meier method will be used to calculate survival (PFS), and the survival comparison of multiple groups will be calculated using the log-rank test. The association between the change in PD-L1 or IFNγ expression and response to treatment will be assessed using a logistic regression model and a multivariate model in which different variables will be included. Pearson’s or Spearman’s correlation test will be used to correlate changes in PD-L1 or IFNγ expression levels in tumor and blood.

## 3. Discussion

Immunotherapy with immune checkpoint inhibitors has improved the prognosis and survival of patients with metastatic malignant melanoma. Its effect is unpredictable and can cause immune-related adverse events in more than a third of the patients. To avoid high grade immune-related adverse events, rapid disease progression or even hyperprogression, new biomarkers are needed to predict response to ICIs.

The human microbiome is the genetic set of all microorganisms in the digestive tract and, by being involved in innate and acquired immunity, influences the effectiveness of immunotherapy treatment. One of the main objectives is to investigate the possibility of using the GIT microbiome as a tumor biomarker. By analyzing the GIT microbiome, it would be possible to identify patients who will benefit from ICI treatment or who will have a higher risk of side effects, even before the start of treatment. In patients with an unfavorable composition of the microbiome, targeted drugs could be used in the first line of treatment in certain cases, thus avoiding the rapid progression of the disease or the side effects of treatment with ICIs. The results of our research will help to more precisely define which phylotypes in the GIT microbiome are more favorable for ICI treatment. Based on this, in the future, the GIT microbiome could also be modified by influencing lifestyle and diet, by adding oral preparations or by fecal transplantation of the microbiome.

PD-L1 expression is variable and changes dynamically during disease evolution; a one-time biopsy sample from tumor tissue is not representative enough. Minimally invasive and real-time detection within peripheral blood exosomes is far more accurate and could provide a better prognostic marker. Interferon Gamma (IFNγ) also correlates with response to ICIs but its detection remains hard to obtain. With liquid biopsy and real-time exosome expression, we could define the patients with higher level of circulating IFNγ and thus build a better response to immunotherapy. Some recent studies have also shown that gastrointestinal microbiome modulates the IFNγ response.

*Our study is the first to combine and investigate multiple possible predictive and prognostic biomarkers at different treatment time periods in first line immune checkpoint inhibitors treatment in metastatic melanoma patients. Outcomes could serve for a better and multi-level understanding of the various factors that influence immunotherapy treatment*.

## Figures and Tables

**Figure 1 biomedicines-11-02016-f001:**
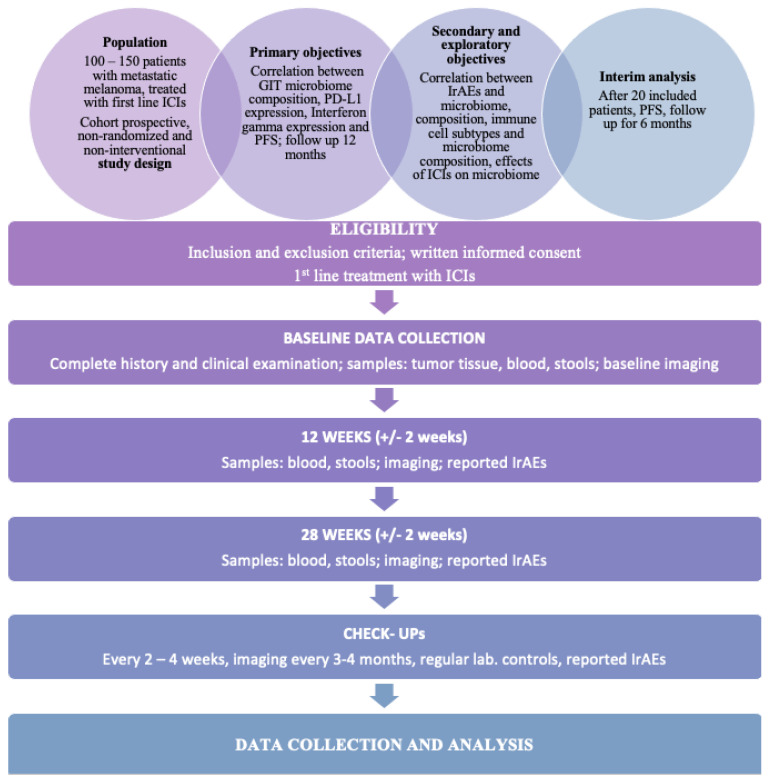
Study flow chart.

**Table 1 biomedicines-11-02016-t001:** List of assessment activities.

	Baseline (4 Weeks or Less before First ICI Application)	Before Every Cycle	12. and 28. Week (+/−2 Weeks)	Progression Disease	Immune-Related Adverse Events
Written consent	X				
BRAF/NRAS status	X				
Clinical examination	X	X		X	X
Personal history	X				X
Laboratory tests	X	X	X	X	X
LDH and S-100	X	X	X	X	
Thyroid function	X	X			X
Immune-related adverse side effects		X	X	X	X
Blood sample	X		X	X	
Stool sample with questionnary	X		X	X	X
CT head, thoracic and abdominal cavity/PET CT + CT head	X		X	X	

**Table 2 biomedicines-11-02016-t002:** Primer details for IFNgama sequence and CD274 (PD-L1) sequence.

Primer	Primer Details	Reference Sequence Number	Assay ID
IFNgama	Expression primer specific for the human IFNgama sequence	NM_000619	Hs.PT.58.3781960
CD274	Expression primer specific for the human CD274 sequence	NR_052005	Hs.PT.58.20819087

**Table 3 biomedicines-11-02016-t003:** Detail description of the used antibodies for flow-cytometric analysis.

Marker	Flurochrome	Clone	Manufacturer	Cat. Number	Volume Used Per 100 µL of Blood Sample/Tube
CD45	V500	HI30	BD	560777	2 µL
CD14	APC-Cy7	MφP9	BD	561709	2 µL
CD68	PE	Y1/82A	Biolegend	BZ-333808	5 µL
CD11b	FITC	Bear-1	Beckman Coulter	IM0530U	10 µL
CD206	PerCP-Cy-5.5	15-2	Biolegend	BZ-321122	5 µL
PD1 (CD279)	PE-Cy7	EH12.1	BD	561272	5 µL
PDL1 (CD274)	APC	MIH1	BD	563741	5 µL
CD4	FITC	SK3	BD	345768	3 µL
CD3	PerCP-Cy5.5	SK7	BD	332771	3 µL
CD8	APC-Cy-7	SK1	BD	348813	3 µL
CD103	PE	Ber-ACT8	Biolegend	350206	5 µL

## Data Availability

Data sharing is not applicable to this article as no datasets were generated or analyzed at this moment.

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
