# Peer review of "The Prognostic and Predictive Value of Human Gastrointestinal Microbiome and Exosomal mRNA Expression of PD-L1 and IFNγ for Immune Checkpoint Inhibitors Response in Metastatic Melanoma Patients: PROTOCOL TRIAL"

_biomedicines, 2023, doi:10.3390/biomedicines11072016_

Round 1

Reviewer 1 Report

In the present study, the authors aimed to evaluate the prognostic and predictive value of gut microbiota and exosomal mRNA expression of PDL1 and INFγ in metastatic melanoma patients under ICIs.

Despite the great interest of the main aim of the study, the results are not supportive enough, mainly due to the way of presentation and english formatting. 

It is advised that the manuscript be written from scratch. 

It is advised that english language be checked from a native english speaker

Author Response

Thank you for your comment. We find your comment unclear, as in this article we are presenting the protocol of the study that is registered and ongoing, and we have no results to present at this moment.

If your comment was refering to the rationale for this study, we added (lines 196-214):

The studies published so far have been carried out on animal models or a small number of subjects or samples. We attribute the conflicting results of the studies conducted so far to this. The importance of intestinal flora in health or pathology and the influence and connection of inflammation and immune cells is currently one of the most current and developing medical topics. It is for this reason that many studies are based on small cohorts and a small number of samples, especially in the field of oncology. Our study is pioneering precisely in this respect, as it is composed thoroughly, with clear current hypotheses, on a large cohort  and number of samples to date. Samples will be taken at three different treatment points and at any event such as hyperprogression, pseudoprogression or immune-related side events. We will study the dynamics of changes in the diversity of the GIT microbiome and the systemic immunological response during the treatment of metastatic melanoma with immunotherapy.

The study is first prospective study that simultaneously determines the expression of IFNƴ and PD-L1 in tumor tissue and blood at different time points. If we demonstrate that there is a correlation in the expression of IFNƴ and PD-L1 in the tumor tissue and the blood, IFNƴ could be obtained from the patient's peripheral blood by a minimally invasive method.

Reviewer 2 Report

Topic is interesting. However, there are some faults and questionable matter ad below.

 1. Each small section of introduction stands alone and does not make sense as a story. Introduction should be reconstructed to show what is unknown matter and what is novelty in this study clearly.

 2. Number of subjects, groups, evaluation indicator should be added to Fig.1. This flow chart has poor information so far.

 3. What is the endpoint to decide dropping out from this study during experiment except for willing of patients? It should be described.

 4. How will immune adverse side effects be evaluated? Percentage of occurrence? Scoring severeness? Evaluation method should be clearly described.

 5. The day when blood samples and stool samples will be collected can be different. Tolerance interval of sampling days should be clear.

 6. Representative garget of molecules of RT-qPCR and ddPCR should be described in tumor tissue analysis and peripheral blood analysis.

7. Consideration of confounding factor to microbiome should be described. Medication except for ICIs is able to affect microbiome.

8. How will relationship between microbiome and beneficial or bad effects of ICIs and immune-related factors including number of peripheral immune cells and IFN-gamma levels be evaluated? Will correlation analysis be performed? Evaluation method should be clearly described.

None.

Author Response

Each small section of introduction stands alone and does not make sense as a story. Introduction should be reconstructed to show what is unknown matter and what is novelty in this study clearly.

Thank you for your comment, we have made additional changes (lines 53-56; lines 92,93 and lines 196-214):

Based on the existing literature, the microbiome, PD-1/PD-L1 signal pathway and Interferon‐gamma (IFN‐γ) are suggested as possible biomarkers, but further studies are needed to obtain relevant clinical information on larger cohorts (lines 53-56).

Human gastrointestinal microbiome,  PD-1/PD-L1 signal pathway and Interferon‐gamma (IFN‐γ) as prognostic and predictive biomarkers (lines 92,93)

The studies published so far have been carried out on animal models or a small number of subjects or samples. We attribute the conflicting results of the studies conducted so far to this. The importance of intestinal flora in health or pathology and the influence and connection of inflammation and immune cells is currently one of the most current and developing medical topics. It is for this reason that many studies are based on small cohorts and a small number of samples, especially in the field of oncology. Our study is pioneering precisely in this respect, as it is composed thoroughly, with clear current hypotheses, on a large cohort  and number of samples to date. Samples will be taken at three different treatment points and at any event such as hyperprogression, pseudoprogression or immune-related side events. We will study the dynamics of changes in the diversity of the GIT microbiome and the systemic immunological response during the treatment of metastatic melanoma with immunotherapy.

The study is first prospective study that simultaneously determines the expression of IFNƴ and PD-L1 in tumor tissue and blood at different time points. If we demonstrate that there is a correlation in the expression of IFNƴ and PD-L1 in the tumor tissue and the blood, IFNƴ could be obtained from the patient's peripheral blood by a minimally invasive method (lines 196-214).

Number of subjects, groups, evaluation indicator should be added to Fig.1. This flow chart has poor information so far.

Thank you, we agree (line 316).

What is the endpoint to decide dropping out from this study during experiment except for willing of patients? It should be described.

Thank you for your comment, we have made additional changes (lines 318-334)

Assesements during the study

Patients will be followed up at their routine visits at the clinic. The follow ups will be scheduled in 3 or 4 week intervals, depending on the ICIs treatment regimen, when routine examination and laboratory test will be taken. Study-specific data will be collected at three different time points: up to 4 weeks before the start of treatment, at 12th (+/- 2 weeks) and 28th week (+/- 2 weeks). Study-specific data will be taken additionaly if suspected disease progression and the appearance of immune-related adverse events or if indicated by oncologist.

Patients will be followed up regularly for the time of the treatment for the first 12 months. The study will be prematurely terminated in an event occurence defined as disease progression/hyperprogression, by a physicians choice in severe immune related adverse events, when high doses of systemic corticosteroid (dexamethasone higher than 4 mg/day or equivalent) or prolonged antibiotic treatment or by the willing of the patient. Before each stool sample donation the patient will fulfill a questionnary regarding eating habits, alcohol and tobacco consumption and medications. In the case of antibotic treatment, the stool sample will be donated after 3 weeks of last antibiotic dose taken.

How will immune adverse side effects be evaluated? Percentage of occurrence? Scoring severeness? Evaluation method should be clearly described.

Thank you, we have added additional explanation (lines 335-345)

Immune-related adverse events (IrAEs) data collection and evaluation

Immune-related adverse events (IrAEs) will be reported at each follow up visits. IrAEs will be evaluated based on the established and standardized grading system Common Terminology Criteria for Adverse Events (CTCAE) developed by the National Cancer Institute. This system grades adverse events based on their severity, ranging from grade 1 (mild) to grade 5 (death). Comprehensive and standardized data on irAEs will be obtained from the patient on each clinical visit, including relevant patient characteristics, timing of irAE onset, clinical manifestations, laboratory findings, imaging results, and interventions undertaken. The biological samples from the patients will be taken if a grade 3 or higher will be noted by a phisician. The biological samples will be taken within 2-weeks from the IrAEs first reported. Results will be given as a percentage of incidence and severity of IrAEs based on grading will also be reported.

The day when blood samples and stool samples will be collected can be different. Tolerance interval of sampling days should be clear.

Thank you, we made additional changes – lines 318-334 (please check the answer for the comment 3)

Study-specific data will be collected at three different time points: up to 4 weeks before the start of treatment, at 12th (+/- 2 weeks) and 28th week (+/- 2 weeks). Study-specific data will be taken additionaly if suspected disease progression and the appearance of immune-related adverse events or if indicated by oncologist.

Table 1: List of assessment activities

Representative garget of molecules of RT-qPCR and ddPCR should be described in tumor tissue analysis and peripheral blood analysis. 

Thank you, we have added additional data (lines 441-470)

For the qRT-PCR assay, SYBR Green chemistry will be used, which enables relative as well as absolute quantification of the target sequence in the investigated sample. Relative quantification will be used to determine the expression level of the transgene in the paraffin samples. Additionally, absolute quantification will be used to determine the target sequences in the plasma of the patients using ddPCR. For both, relative and absolute quantification we will use specific set of primers (PrimeTime qPCR Primers, IDT Technologies), that do not amplify any other known dsDNA sequence, and are specific for the IFNgama sequence and CD274 (PD-L1) sequence (primer details are in Table 2). 

  • Sample preparation for flow-cytometric measurement will be carried out as previously described by our group (40, 41). Used antibodies (Table 3) will be divided into 2 separate tubes according to the analyzed immune cells, and half a million cells per 100 μl were used for each tube. Immune cells will be gated according to their immunophenotype: T cells (CD3+), helper T cell subset (CD4+), cytotoxic T cell subset (CD8+), and macrophages (CD11b+CD14+) and their M1-like (CD86+) and M2-like (CD206+) subsets. Percentages of T cells and macrophages will be calculated as a ratio per CD45+. Percentages of CD4+ and CD8+ T cell subsets will be given as a percentage of CD3+, while M1-like and M2-like macrophage subsets were given as a percentage per all macrophages. Expression of PD-1 and PD-L1 will be analyzed on both T cells and macrophages, as well as on their corresponding subsets. The correlation of the analyzed immune cells and their subsets with patient PFS and OS will be also calculated. Survival analysis will be based on a X-year patient follow-up. Flow-cytometric data will be acquired with a 10-color BD FACSCanto™ II Flow Cytometer and FACSDiva 8.0.2 software (BD Bioscience, USA). FSC files will be analyzed using FlowJo v10.8 1 (BD Biosciences, USA).

Table 2: Primer details for IFNgama sequence and CD274 (PD-L1) sequence

Table 3: Detail description of the used antibodies for flow-cytometric analysis

Consideration of confounding factor to microbiome should be described. Medication except for ICIs is able to affect microbiome.

Thank you for your comment. Patients on mediactions possibly affecting microbiome would not be included.

How will relationship between microbiome and beneficial or bad effects of ICIs and immune-related factors including number of peripheral immune cells and IFN-gamma levels be evaluated? Will correlation analysis be performed? Evaluation method should be clearly described.

Thank you, we have made this more clear (lines 488-490)

The gastrointestinal microbiome, PD-L1 and Interferon Gamma expression will be correlated with one year PFS as the primary outcomes of the study. An interim analysis will be made after first 20 included patients.

Round 2

Reviewer 1 Report

Thank you for the reply

Reviewer 2 Report

The manuscript has been revised well.

fine